# Simultaneous Degradation Study of Isomers in Human Plasma by HPLC-MS/MS and Application of LEDA Algorithm for Their Characterization

**DOI:** 10.3390/ijms232113139

**Published:** 2022-10-28

**Authors:** Marco Pallecchi, Laura Braconi, Marta Menicatti, Sara Giachetti, Silvia Dei, Elisabetta Teodori, Gianluca Bartolucci

**Affiliations:** 1NEUROFARBA Department, Section of Pharmaceutical and Nutraceutical Sciences, University of Florence, Polo Scientifico, Via U. Schiff 6, Sesto Fiorentino, 50019 Firenze, Italy; 2Chemistry Department DICUS ‘Ugo Schiff’, University of Florence, Polo Scientifico, Via della Lastruccia 3, Sesto Fiorentino, 50019 Firenze, Italy

**Keywords:** triple quadrupole system, ERMS, breakdown curves, collision energy, enzymatic degradation kinetics

## Abstract

This paper proposes a tandem mass spectrometry (MS/MS) approach in isomer recognition by playing in the “energetic dimension” of the experiment. The chromatographic set up (HPLC) was tuned to minimize the run time, without requiring high efficiency or resolution between the isomers. Then, the MS/MS properties were explored to solve the signal assignment by performing a series of energy resolved experiments in order to optimize the parameters, and by applying an interesting post-processing data elaboration tool (LEDA). The reliability of the new approach was evaluated, determining the accuracy and precision of the quantitative results through analysis of the isomer mixture solutions. Next, the proposed method was applied in a chemical stability study of human plasma samples through the simultaneous addition of a pair of isomers. In the studied case, only one of the isomers suffered of enzymatic hydrolysis; therefore, the influence of the stable isomer on the degradation rate of the other was verified. In order to monitor this process correctly, it must be possible to distinguish each isomer present in the sample, quantify it, and plot its degradation profile. The reported results demonstrated the effectiveness of the LEDA algorithm in separating the isomers, without chromatographic resolution, and monitoring their behavior in human plasma samples.

## 1. Introduction

Most medicinal chemistry investigations involve the design, chemical synthesis and development of pharmaceutical agents or bio-active molecules (drugs) suitable for therapeutic use [1]. These studies produce a large number of compounds that must be qualified and tested to evaluate their characteristics and potential applications as new candidate drugs. Therefore, a series of screening experiments will be scheduled to select the best candidate molecules that will continue in their development as pharmaceutical agents. For this reason, it will be necessary to arrange specific, reliable, fast and relatively cheap analytical methods to support this research [2]. Mass spectrometry (MS) is an attractive technology for this purpose, thanks to its high selectivity, wide dynamic range and high throughput capabilities [3]. Indeed, it is widely used in analytical methods to obtain quali-quantitative information on the analytes [4,5]. However, the specificity of the mass spectrometry signal may be compromised when the compounds show both common fragment ions and similar chromatographic retention properties, as in the case in isomers analysis. To expand the MS application, some have authors explored the possibility of recognizing isomers by tandem mass spectrometry (MS/MS) experiments, based on the intensity of “diagnostic fragments” [6]. Unfortunately these cases rarely occur, while it is common to obtain a MS or MS/MS spectra of isomers with the same ion signals and, even if different branching ratios are present, it is hard to distinguish them if simultaneously present in the sample.. In the simplest scenario, when similarly fragmenting isomers are present in a sample, it is common practice to separate them chromatographically to allow their quali-quantitative characterization [7,8]. Generally, this approach may require a longer time to set up the separation method with the evaluation of different chromatographic columns, mobile phases and elution programs to obtain adequate analytes separation [9,10]. All of these procedures are usually molecule-specific and can rarely be extended to other compounds. Nevertheless, in the last two decades, many MS/MS strategies have been developed to solve this problem by allowing the characterization and quantification of isomers and/or isobars in mixtures via a standardized approach, applicable to different compounds [11]. Among these strategies, the most promising and interesting for a widespread application in the recognition of isomers are based on (1) energy-resolved tandem mass experiments [12,13,14,15,16,17,18,19] and (2) kinetics of the ion-molecule interaction [20,21,22], even if the latter are limited to the use of the ion trap. Summarizing the reported results, the discrimination between isomers was achieved by optimizing the selection of precursor ion, its fragmentation through collision induced dissociation (CID) mechanism and the analysis of fragmented ions produced. Each of these phases were explored, developed and tuned to carry out an adequate specificity to distinguish the isomers in the sample, without the support of any structural manipulation (i.e., derivatization and isotopic enrichment) or chromatographic separation. The use of the right approach provides many analytical advantages, among which the most important are sensitivity, reliability and faster analysis.

In regards the topic of the recognition of isomers, our group proposed and developed an MS/MS post-processing mathematical algorithm named LEDA (Linear Equation of Deconvolution Analysis) that allows the recognition of isomer compounds without their chromatographic separation [23,24,25,26]. Profiting by LEDA features, recently we introduced a methodological approach that simplifies the liquid chromatography (HPLC) parameters, allowing the use of a short column and a fast elution gradient, leading to increased productivity, without losing determination specificity [27]. Within this approach, the chromatographic column was used only to avoid or limit the interference of the sample matrix towards the analyte ionization process (matrix effects).

Thanks to this experience, the logical trend suggests the evaluation of the hydrolytic kinetics of plasma enzyme systems when spiking a pair of isomers in the sample. In detail, we propose the case where only one of the isomers is hydrolyzed; in this way, adding the sample to the mixture of isomers, we can verify the influence of the stable isomer on the degradation rate of the other.

To carry out this evaluation, the LEDA approach was applied to the plasma stability experiments on two pairs of isomers of P-glycoprotein (P-gp) inhibitors candidate. These molecules are characterized by a tertiary amine group carrying two polymethylene chains of variable length (between 3 to 6 carbon atoms), linked to different aromatic moieties through two ester bonds [28]. The structures of the studied compounds are reported in Figure 1. The panel of P-gp inhibitors is represented by positional isomers with different lengths of the hydrocarbon chains linked to the aromatic moieties; then, a proper analytical method for each couple of isomers with suitable detection selectivity is required to avoid any mutual interferences.

The aim of this work is to propose a single HPLC separation method and to develop a series of MS/MS conditions that allow the recognition of studied isomer pairs when they are present simultaneously in the sample, supporting their chemical stability study.

## 2. Results

The investigation was devoted to monitoring the hydrolytic activity of human plasma enzymes towards selected couples of isomers simultaneously added to the samples. The description of the experimental results of the proposed study is presented in the following steps:the checking of the achieved chromatographic separation;the exploration of the MS/MS features in the isomers distinction;the evaluation of a mathematical device (LEDA) that allows the conversion of the common MS/MS signals in specific isomer abundances, related to their relative concentration in the sample;the assessment of the LEDA quantitative performances;the application of the optimized conditions of the LEDA approach in plasma stability experiments.

### 2.1. Chromatographic Separation

The obtained chromatographic profiles showed an appropriate distinction between the internal standard (IS) and the analytes as a result of the different characteristic ion transitions (MRM signals), however, an acceptable separation among the isomers was not reached. The peak parameters (i.e., retention times, peaks width, efficiency, etc.) for each analyte were calculated and reported in the Appendix A. In Figure 2, an example of comparison between the HPLC-MS/MS analysis of the ELF94 and ELF96 isomers is shown.

The chromatographic profiles of the studied compounds are reported in the Appendix A.

The proposed HPLC method achieved a poor efficiency, evaluated as the number of theoretical plates (*N* ≈ 5000), mainly due to the short column used. In any case, this set up allowed us to take advantage of a rapid elution gradient, allowing for a brief window to rinse the column and restore the initial condition; as a result, a faster chromatographic run is obtained. Therefore, the primary feature required by chromatography in the presented approach was reached: being simple and rapid. In this way, it was possible to process a large number of samples (about 150–160/day) with the same reliability of common HPLC-MS/MS methods, avoiding the complicated method setup for each pair of isomers. However, to obtain a suitable distinction (analytical specificity) between the tested compounds and to ensure their proper monitoring during the degradation experiment, the MS/MS features for each analyte must be explored.

### 2.2. Collision-Induced Dissociation Sudy

The studied compounds (see Figure 1) underwent electrospray ionization (ESI) in positive ions MS conditions, and showed only an abundant signal of protonated molecules ([M + H]^+^ species), with the same *m*/*z* value for each pair of isomers (Appendix A). Consequently, the advantages of tandem mass spectrometry experiments for obtaining the characteristic signals, and for allowing the recognition of isomers, were examined. To highlight the possible intrinsic differences in the molecular stability of the isomers, a series of fragmentation studies on the [M + H]^+^ species, at different collision voltages (CVs), were performed (details in Section 4.4). The set of MS/MS acquisitions at different CVs is referred to as energy resolved mass spectrometry or ERMS experiments. By processing the data obtained from the ERMS experiments, the collision breakdown curves were plotted and the most significant product ions (Pis) for each couple of isomers were selected (Appendix A) to set up a proper acquisition MRM method (Section 4.4). A typical comparison between the collision breakdown curves of FRA76 and GDE5 isomers is reported in Figure 3.

Based on the collision breakdown curves of each couple of isomers, the six most representative Pis (>10%fragmentation yield) could be selected. Four of these Pis were related to the bond cleavages of the ester groups, following the general scheme proposed in Figure 4.

The MS/MS most intense signals, for all compounds, were obtained by the bonds cleavage 1 and/or 2 with the formation of Pi_1_ and/or Pi_2_ and the corresponding neutral alkyl-alcohol structure. Pi_1_ and Pi_2_ (*m*/*z* 221 or 195, respectively) show the same structure regardless of the precursor compound, but their maximum abundances are different and characteristic for each isomer (Appendix A). On the other hand, it was interesting to note that the ion structures of the Pi_3_ and Pi_4_ maintained the information about the differences between the isomers and would have been useful to their distinction. Unfortunately, these Pis show a poor formation yield (<20%), reducing their ability for the recognition of isomers. Finally, two other Pis were observed (Pi_5_ and Pi_6_) that can be related to the bond cleavage between the tertiary amine group and the propyl chain linked to the aromatic moiety (Ar_1_ or Ar_2_), but their formation yield is also less than 20%. In addition, in this case, the structures of the ions are the same, regardless the considered isomer, but their relative abundances are different. The fragmentation hypothesis and molecular structures of Pi_5_ and Pi_6_ are reported in the Appendix A.

The fragmentation studies on the [M + H]^+^ species of the selected P-gp inhibitors highlighted common dissociation pathways with the formation of six main Pis that could characterize each isomer. Indeed, these ions show different abundances depending on the precursor isomer; hence, it may be possible to set up a suitable “mathematical device”, able to distinguish the common signals and elaborate the MS/MS data by assigning the proper abundance to the isomer or isomers present in the sample.

### 2.3. LEDA Algorithm

The proposed mathematical device, LEDA, is a matrix of linear equations that elaborates each MS/MS signal, distinguishes the possible components and assigns the correct abundance to the isomers present. The algorithm is based on the rationale that the MS/MS signal might be represented as the sum of contribution of each isomer present in the sample. As demonstrated in the breakdown curves shown above (Figure 3), each isomer produces the same panel of Pis, but with different formation yield. Therefore, it is reasonable to describe each Pi signal as the sum of abundances of this ion deriving from all of the isomers that are eventually present in the sample. To manage reliable data and to enhance the compound-dependent Pi yield differences, the relative abundances of selected Pis with respect to a reference ion (Ri) were calculated. For this purpose, the precursor ion signal was acquired as Ri at the CV value corresponding to its highest intensity in the breakdown curve, without any fragmentation process occurring. Through this approach, the ratio between the abundance of each Pi acquired and the abundance of Ri represents the yield of the Pi formation at the selected CV. The CVs applied for Ri and Pis ion transitions selected in this study were reported in Section 4.4. Therefore, knowing the characteristic abundance ratios of pure isomer (Section 4.8), a deconvolution of acquired MS/MS signals is possible based on a series of linear regression equations as follows:(1)$${\left(\frac{\mathrm{Pi}}{\mathrm{Ri}}\right)}_{m}=\overset{n}{\underset{x\;=1}{\sum }}{\left(\frac{\mathrm{Pi}}{\mathrm{Ri}}\right)}_{x}\times {\left[\%\right]}_{x}$$

(Pi/Ri)_m_ is the abundance ratio between the product ion (Pi) vs reference ion (Ri) measured (m) in the sample.(Pi/Ri)_x_ is the characteristic abundance ratio between the Pi vs Ri of pure isomer.[%]_x_ is the concentration (%) of the isomer in the sample.

Considering a binary mixture of isomers (A—B), a single Equation (1) related to only a product ion ratio (Pi/Ri) could be sufficient. Indeed, by assuming that only the pair of isomers constitutes the MS/MS signal, the concentration of B is calculated as B% = (1 − A)%. However, in this case the possible contribution of signals from unknown isomers (or any co-eluting compound having the same product and reference ions) is neglected. Therefore, to avoid ill-conditioned algorithms, the resolution of the mixtures of *n* isomers requires a matrix composed of *n* linear regression Equation (1) obtained from *n* different experimental data. In our case, the experimental data used were the abundance ratios of product/reference ions selected during the MS/MS method set-up. In this study, a pair of isomers was spiked in the samples; then, a LEDA matrix composed by two equations (square matrix) could be adequate. However, to verify the reliability of MS/MS signal deconvolution, an overdetermined system of linear equations (six equations) was also evaluated by assembling a LEDA matrix that includes all the Pis revealed during the ERMS experiments. Therefore, two LEDA matrices have been proposed: the first was composed by only two equations (LEDA_195-221_, Appendix A), related to the data from most abundant fragmented ions, Pi_1_ and Pi_2_ (*m*/*z* 221 and 195, respectively); the second was (LEDA_all_, Appendix A) assembled by all the Pis acquired.

### 2.4. LEDA Reliability

The LEDA reliability was evaluated by determining the accuracy and precision of the quantitative results obtained from processing the MS/MS data from the analyzes of the mixture solutions (Section 4.3) of each pair of isomers. The algorithm elaborated the acquired MRM signals, separating their components and assigning the correct abundance to the identified isomers. In the present case, the integrated area of the chromatographic peak, corresponding to the elution of both isomers, was elaborated, the MS/MS signal assigned, and the relative abundance of each component calculated (peak purity %). Since five standard mixtures with different isomer concentrations were involved in the evaluation of the performance of the LEDA algorithm, a tool should be introduced that describes the accuracy and precision of the method considering all the processed solutions. To this end, a graphical distribution of the results calculated by the LEDA algorithm was carried out with respect to the expected concentration values. Then, a linear regression analysis was performed to find the best fitting between the data points (validation plot). The calculated values of the linear function (i.e., slope, y-intercept, etc.) represent the quali-quantitative reliability characteristics of the LEDA approach. The accuracy of the method is related to the slope of calculated linear function; indeed, by assuming that LEDA results are consistent with expected values (y ≈ x values), the obtained linear function shows a slope close to 1 (100% accuracy). Naturally, in case of experimental data, the plotted points are casually distributed around the calculated linear function; then the standard deviation of residues (error standard of linear function) could represent the precision of data processing. Finally, the y-intercept and the determination coefficient describe the minimal concentration of analyte recognizable in mixture and the confidence of the estimated function to represent the data distribution, respectively. Thus, by introducing the validation plot, it was possible to estimate the accuracy or precision of the LEDA processing with a single value representative of all the concentration levels tested. In Figure 5, an example of a validation plot obtained for the ELF96 isomer by processing the standard mixtures with LEDA_195–221_ matrix is shown.

The reported validation plot demonstrated how the estimated linear function was representative of the data points (R^2^ = 0.999), the slope near to 1 indicates an optimal accuracy of recognition of the ELF96 isomer in the analyzed mixtures and the y-intercept close to the origin assures a good isomer distinction also in low relative abundance. Clearly, the minimum amount of isomer detectable in the mixture also depends on the precision of LEDA elaboration, represented by the standard error of the calculated linear function (SE-Lin). Then, to estimate the best accuracy and precision performances of the LEDA process, two mathematical matrices for each pair of isomers were used to elaborate the MS/MS data acquired from the analyzes of the standard mixtures. The comparison with obtained results were reported in Table 1, while the validation plots of all the analytes were collected in the Appendix A.

All the slope values shown in Table 1 are close to 1, which represents a recognition accuracy near to 100%, confirming the reliability of all employed LEDA matrices to distinguish the correct amount of each isomer present in standard mixtures. Moreover, the precision features (estimated by SE-Lin values) were interesting; indeed, the purity ratio values ranged between 0.01 to 0.03, corresponding to an uncertainty of the 1% and 3% of the isomer present in mixture, respectively. Then, together with the y-intercept values near the origin, the assembled LEDA algorithms achieved a correct assignment of the MS/MS signal up to 2–6% of relative abundance of the isomer in the mixture (considering the 95% confidence interval such as ± 2 SE-Lin). The agreement between the results obtained by the used LEDA matrices (LEDA_195–221_ vs LEDA_all_) that showed no significant differences was noteworthy. In this case, the evidence suggests that increasing the number of Equation (1) to compile the LEDA matrix does not influence the accuracy and precision of the elaboration. However, the outcome could depend by the low number of isomers involved (only two) or by the “simplicity” of the standard mixtures, which were diluted in neat solvents. To settle this point, both LEDA matrices were applied in plasma stability studies of the selected couple of isomers to also verify the effectiveness of the algorithm in a complex matrix.

### 2.5. Evaluation of the Quantitative Performance of the MS/MS Analysis

In this section, the parameters (linearity, limit of detection, limit of quantitation, matrix effects and recovery) that define the range of applicability and the trueness of the quantitative analysis by the proposed approach are reported and described.

The analyte determination was carried out by using the verapamil hydrochloride as internal standard (IS) that, with its similar molecular characteristics to the studied compounds, assures reliable and robust quantitative data analysis. The linearity range of the of response of each compound was verified by analyzing, with a proper MS/MS method, its calibration solution (Section 4.3) and plotting the respective calibration curves. As described in Section 4.7, these curves were built by using both the MRM transition related to the precursor ion (used also as Ri signal) and the most abundant Pi present in its MRM spectrum (Appendix A) as quantitative signals for each analyte. Thus, each analyte reports two calibration curves (corresponding to precursor or Pi signals) that will be useful in the evaluation and application of the LEDA tool for the determination of the isomer. The obtained linear regression parameters, the determination coefficient (R^2^) and the estimated LOD and LOQ values for each analyte are reported in the Appendix A. It is worth emphasizing that the slope values of the calibration curves referring to the Pi signals of each isomer are significantly different. This discrepancy is a consequence of both the different quantitation Pi signal considered for each isomer and its different yield of formation. On the contrary, the slope values obtained from the precursor ion transitions are similar between each pair of isomers. Since the slope value of the calibration curve represents the quantitation sensitivity of the analyte [26], by comparing the reported slope values, it can obtain similar sensitivities for both isomers by using the precursor ion curves. Naturally, this observation does not consider background noise, isobaric interferences, or other factors due to the sample matrix that effect the precursor ion signal, compromising its value. However, the use of the LEDA algorithm solved these problems by processing the acquired MS/MS data and separating the defined components that may be present from the non-specific signal. Therefore, it is possible to allocate the correct percentage of the MS/MS precursor ion signal for each isomer, then to estimate the ratio values with the abundance of IS (peak area ratios or PAR’s), and finally to calculate their concentration by respective calibration curves. The proposed approach not only allows the use of the higher calibration curve slope, but also has the same quantitative relevance, as shown by the similar slope values obtained from the isomer precursor ion data. Therefore, by using these calibration curves, it is possible to have the same quantitative sensitivity for both analytes, neglecting the signal loss due to the formation yield of the product ions.

The standard error of y-intercept (Y-SE) of linear regression, calculated for each calibration curve, was used to establish the limit of detection (LOD) and limit of quantitation (LOQ) of the analytes in the selected quantitation signals (Section 4.7). By following the proposed method, the LOD and LOQ values of each compound were calculated on the Y-SE of quantitation ion signal, neglecting either the absolute intensity value or the evaluation of the background noise, which can be variable and dependent upon several factors. In this way, it was possible to obtain LOD and LOQ values each time the calibration curve was performed, enabling the monitoring of the instrumental performances between different analysis batches. Moreover, the obtained LOD and LOQ values strengthened the reliability of the low concentration levels chosen for the calibration curves, allowing for the analyte detection in the plasma samples less than 10% (<0.1 μM) as compared with the spiked concentration (1 μM).

The matrix effects (ME) and recovery (RE) values for IS and each analyte were obtained through the analysis of the sample sets prepared, as described in Section 4.6, using the proposed HPLC-MS/MS methods. The acquired data were processed, as reported in Section 4.7, and the results are shown in the Appendix A. The ME values (ranging between 88% and 101%) demonstrate that the signal abundance of each compound was not significantly different both in the human plasma (set B) and in the neat solvents (set A) samples. Therefore, the chromatographic approach (short column and fast gradient elution) was suitable to avoid the harmful effect of the plasma matrix on the analyte ionization efficiency. The RE ratio between the data obtained from set C and set B describes the recovery of the analyte during the sample preparation; since the obtained values vary between 87% and 99%, it was demonstrated that the analytes were correctly extracted from the plasma matrix. Finally, the obtained ME and RE values for the IS resulted about 100% for the tested matrix, confirming that it was an appropriate choice to monitor the studied compounds.

The collected results about quantitative parameters demonstrated that the proposed HPLC-MS/MS methods are suitable to be applied in the chemical stability study of selected P-gp inhibitors.

### 2.6. Chemical Stability Test

The present investigation was carried out to evaluate the variation of hydrolytic activity of the human plasma enzymatic system when two potential substrates were simultaneously present. To achieve this aim, we studied the behavior of two couples of isomers which, in a previous study [28], had been shown to undergo enzyme hydrolytic activity with an isomer, while the other remained stable. Therefore, the human plasma sample solutions, prepared with a mixture of isomers, presented both a substrate and a potential interference, which could influence the enzyme functionality. To correctly monitor this process, it must be possible to distinguish each isomer present in the sample, quantify it with the proper calibration curve and plot its degradation profile. The LEDA algorithm was used to support this investigation and to test its quali-quantitative ability in a complex matrix of human plasma samples. The experimental plan involved the analysis of three series of human plasma samples for each couple of isomers: two series of samples were added to a pure isomer, to check the activity of the plasma enzyme system without interferences, and the last series of samples were spiked with the equimolar mixture of isomers. The series of samples were prepared, as described in Section 4.6 and analyzed with the proper HPLC-MS/MS method. The acquired data were processed with the conventional quantitative method for the series containing the single isomers, and with the LEDA approach for the samples spiked with the mixture of isomers. The samples containing a single isomer were worked out by the integration of quantitative Pi signal and the estimation of PAR value by dividing with IS abundance, and quantitative determination of the analyte in the sample was obtained by using the proper calibration curve. In contrast, in the samples spiking with the mixture of isomers, each proposed LEDA matrix recognized the relative concentration of the isomers present in the sample. Therefore, their abundance in the precursor ion signal (or Ri signal) were assigned, each PAR value calculated by dividing with the IS abundance, and quantitative determination of the analytes in the sample were performed by using the respective precursor ion calibration curve. In Figure 6, an example of the degradation plots of the mixtures of the ELF94 and ELF96 isomers is shown, while all the degradation profiles are reported in the Appendix A.

The characteristic degradation parameters (half time or t_1/2_ and degradation rate or k) of the samples added with the mixture of isomers were calculated by processing the quantitation data obtained from the proposed LEDA matrices and the results, reported in the Appendix A, were compared. In addition, the tested LEDA matrices showed equivalent results, despite the complexity of the sample matrix analyzed. This evidence established that the LEDA_19–-221_ matrix assembled with two Equation (1), relating of the most intense Pis (Pi_1_ and Pi_2_) in the isomers fragmentation, was adequate to represent the system of the studied compounds, ensuring their correct distinction in sample mixtures. Therefore, all the data reported below were obtained by processing the MS/MS data with the LEDA_195–221_ matrix.

The effectiveness of the LEDA approach was evaluated by comparing t_1/2_ and k, calculated from the analyzes of the three sets of plasma samples (Table 2). The obtained results confirmed the hydrolysis of substrate-isomers (FRA76 and ELF94) by plasma enzymes, while the concentration of stable isomers (GDE5 and ELF96) remained constant in all sample sets tested. The comparison of the degradation parameters (t_1/2_ and k values), reported in Table 2, highlights some differences between the absolute values of the data obtained from the sets of samples (i.e., FRA76 t_1/2_ 43 and 32 min.), however, these differences are statistically non-significant (considering the error values, the confidence range of the measurements overlaps).

To verify the efficiency of the LEDA algorithm, the set of plasma samples spiked with an equimolar mixture of isomers was processed using the ‘scan-by-scan’ method, allowing the graphical separation of the isomers present in the chromatographic profiles. In this case, the computation of the LEDA matrix was repeated on each MS/MS data point acquired; consequently, a relative abundance of the Ri signal for each isomer was assigned. By considering a generic chromatographic profile, the performance of the algorithm was reflected in the deconvolution of unresolved peak of Ri signal that delivered its abundance between the isomers (LEDA reconstructed chromatographic profiles). In Figure 7, an example of LEDA reconstructed chromatographic profiles of the human plasma sample spiked with the mixture of the ELF94 and ELF96 isomers at the incubation time of 0 min was shown.

In Figure 7, the reconstructed profiles showed an equal abundance of the isomers in the Ri signal, with the mixture solution added, verifying that at the incubation time of 0 min, the plasma enzymes did not work. Increasing the incubation times, the reconstructed profiles of substrate-isomer (in the case ELF94) reported a decrease of the peak intensity, while the peak of the stable isomer (in the case ELF96) maintained the same abundance (Appendix A). Therefore, the LEDA ‘scan-by-scan’ processing method confirmed the data obtained in the previous elaboration (peak area integration) and demonstrated its reliability by distinguishing the abundance of each isomer present in the unresolved Ri signal. The LEDA-reconstructed chromatographic profiles of all the sample involved in this study are reported in the Appendix A.

## 3. Discussion

In this study, we proposed the evaluation of the activity of hydrolytic human plasma enzymes towards the simultaneous presence of two possible substrate compounds. To achieve this aim, we selected two couples of isomers, one pair was characterized by a substrate isomer (susceptible to hydrolysis), while the other remained stable. The challenge of this application was to monitor the behavior of both isomers when they were present at the same time in the sample by using an HPLC-MS/MS method. Indeed, the MS and MS/MS methods are known to have poor specificity in the recognition of isomers. To overcome these difficulties, a new HPLC-MS/MS approach was proposed, using an interesting post-processing data elaboration (LEDA) to solve possible signal misassignment. Taking advantage of the characteristics of LEDA, the chromatographic set up to analyze the samples was tuned to minimize the run time, without requiring high efficiency or resolution between the analytes. The only aim of the chromatographic column was to avoid or limit the interference of sample matrix in the analyte ionization process (matrix effects). Therefore, we were able to develop an HPLC-MS/MS quantitative method that is sensible, specific and fast.

The LEDA “mathematical device”, introduced to process the acquired MS/MS data, was demonstrated to be reliable in recognizing and separating the possible components present in the sample signals. Its effectiveness was tested by processing the HPLC-MS/MS analysis of a series of known mixture solutions, and we introduced the “validation plot” that, through the evaluation of the characteristic parameters, described the accuracy and precision of the quantitative ability of the LEDA elaboration. The obtained results showed an accuracy near to 100%, with precision values ranging between of 1% to 3%. The obtained MEs values (ME 88–100%) demonstrated that the chromatographic setup (short column and fast elution gradient) was required to avoid the matrix interferences, while the REs values (RE 87–99%) indicate a reliable sample preparation, despite only a protein precipitation being carried out. Therefore, the general procedure proposed was found to be adequate in studying the selected isomer compounds without their chromatographic separation and applying and developing the MS/MS features. For this purpose, the LEDA post-processing algorithm was introduced to allow the use of the sole and simple chromatographic method, which led to increased productivity, without losing determination specificity.

Finally, we applied the developed HPLC-MS/MS method to the analysis of three series of human plasma samples, spiked with each pure isomer (series 1 and 2), and an equimolar mixture of both isomers (series 3). The obtained results confirmed the stability characteristics of the selected compounds: the substrate isomers were hydrolyzed, while the other isomers kept their concentration constant in the sample solution. Furthermore, the presence of the stable isomer did not appear to affect the degradation of the substrate-isomer (no significant differences between the t_1/2_ e k rate).

## 4. Materials and Methods

### 4.1. Chemicals

The P-gp inhibitors selected for this study were obtained as reported in Ref. [27]. Acetonitrile (Chromasolv), formic acid, ammonium formate (MS grade), NaCl, KCl, Na_2_HPO_4_ 2H_2_O, KH_2_PO_4_ (reagent grade), verapamil hydrochloride and ketoprofen (racemic analytical standard) were purchased by Merck (Milan, Italy). Ketoprofen ethyl ester (KEE) was obtained by Fisher’s reaction from ketoprofen and ethanol. Ultrapure water or mQ water (resistivity 18 MΩ cm) was obtained from Millipore’s Simplicity system (Milan, Italy). Phosphate buffer solution (PBS) was prepared by being dissolved in mQ water and reported salts at the following concentrations: 8.01 g L^−1^ of NaCl, 0.2 g L^−1^ of KCl, 1.78 g L^−1^ of Na_2_HPO_4_ 2H_2_O, and 0.27 g L^−1^ of KH_2_PO_4_. The samples of human plasma were collected from healthy volunteers, pooled and kept at −80 °C until use. All subjects gave their informed consent for inclusion before they participated in the study. The study was conducted in accordance with the Declaration of Helsinki, and the protocol was approved by the local Ethics Committee (Comitato Etico Regionale per la Sperimentazione Clinica della Regione Toscana, Sezione AREA VASTA CENTRO) Institutional Review Board (CE: 11166_spe, 11 September 2018 and CE: 10443_oss, 14 February 2017).

### 4.2. Instrumental

The HPLC-MS/MS analysis was carried out using a Varian 1200 L triple quadrupole system (Palo Alto, CA, USA), equipped with two Prostar 210 pumps, a Prostar 410 autosampler and an electrospray ion source (ESI). Raw data were collected and processed by Varian Workstation version 6.8 software (Palo Alto, CA, USA). A thermostatic oven G-Therm 015 (F.lli Galli, Milan, Italy) was used to maintain the samples at 37 °C during the degradation test, while the centrifuge Eppendorf 5415D (Merck, Milan, Italy) was employed to centrifuge the plasma samples.

### 4.3. Standard and CalibrationSsolutions

Stock solutions of analytes and verapamil hydrochloride (internal standard or IS) were prepared in acetonitrile at 1.0 mg mL^−1^ and stored at 4 °C. Working solutions of each analyte were freshly prepared by diluting stock solutions up to a concentration of 1.0 µg mL^−1^ and 0.1 µg mL^−1^ (working solutions 1 and 2, respectively) in mixture of mQ water:acetonitrile 50:50 (*v*/*v*). The IS working solution was prepared in acetonitrile at 33 ng mL^−1^ (IS solution). The quantitative data of each analyte were obtained by building a five-level calibration curve, prepared by adding proper volumes of the working solution (1 or 2) of each analyte to 300 μL of IS solution. The obtained solutions were dried under a gentle nitrogen stream and dissolved in 1.0 mL of 10 mM of formic acid in mQ water:acetonitrile 80:20 (*v*/*v*) solution. Final concentrations of calibration levels of each analyte were: 5.0, 10.0, 25.0, 50.0, and 100.0 ng mL^−1^. With the aim of evaluating the accuracy and precision of the proposed approach, five mixture solutions for each couple of isomers in different proportions were prepared. Mixture 1 had 90% of A isomer and 10% of B isomer, while in mixture 2 the proportion was 75% and 25%; mixture 3, 50% and 50%; mixture 4, 25% and 75%; and mixture 5, 10% and 90%. The sum of isomer concentrations in the five mixture solutions was 100 ng L^−1^. Each solution was used to prepare the precision and accuracy set samples as follows: 100 μL of mixture solution and 300 μL of IS solution in autosampler vial were added. The obtained solutions were dried under a gentle nitrogen stream and dissolved in 1.0 mL of 10 mM of formic acid in mQ water:acetonitrile 80:20 (*v*/*v*) solution. The calibration levels and mixture solutions were analyzed six times by the HPLC-MS/MS system, with the proper conditions. Finally, the spiked solutions, used in the sample preparation of chemical stability study, were separately prepared by diluting the respective stock solutions in mQ water:acetonitrile 80:20 (*v*/*v*) solution, to obtain an analyte final concentration of 10 μM. For each couple of isomers, three spiked solutions were prepared: solution of isomer A, solution of isomer B and solution of mixture of both isomers (each at concentration of 10 μM).

### 4.4. HPLC-MS/MS Methods

The chromatographic parameters employed to analyze the samples were tuned to minimize the run time. The column used was a Luna C18 20 mm length, 2 mm internal diameter and 3 μm particle size Phenomenex (Torrance, CA, USA), at a constant flow of 0.25 mL min^−1^, employing a binary mobile phases elution gradient. The solvents used were 10 mM formic acid and 5 mM ammonium formate in mQ water:acetonitrile 90:10 (solvent A) and 10 mM formic acid and 5 mM ammonium formate in acetonitrile:mQ water 90:10 (solvent B). The program of elution gradient was set up as follows: initially at 90% solvent A, which was then decreased to 10% in 4.0 min, kept for 2.0 min, returned to initial conditions in 0.1 min and maintained for 2.0 min to a total run time of 8.0 min. The column temperature was maintained at 40 °C and the injection volume was 5 µL. The ESI source operated in positive ion mode by using the following setting: 5kV needle, 42 psi nebulizing gas, 600 V shield, and 20 psi drying gas at 280 °C. The MS analyzes were obtained in an ion scan by acquiring the *m/z* range from 250 to 750 with 600 ms of scan time. The ERMS experiments were performed to study the fragmentation of [M + H]^+^ species of each analyte and build its breakdown curves [23,24,25,26,27]. The ERMS experiments were carried out by a series of product ion scan (MS/MS) analyses, increasing the CV stepwise in the range 5–50 V. Each acquired MS/MS spectrum was in the *m/z* range from 50 to 650, scan time of 600 ms and by using argon as collision gas. The ERMS experiments were performed by introducing working solution 1 to each analyte (Section 4.3), via syringe pump at 10 µL min^−1^; the protonated molecule was isolated, and the abundance of product ions were monitored. The obtained data were used to build the breakdown curves that describe the fragmentation of the precursor ion related to the CV applied. The breakdown curves were plotted by reporting the ratio between the values of intensity (averaging about 15–20 scans) of each ion signal, present in the MS/MS spectra acquired, versus the maximum intensity of precursor ion (Ri) and the CV applied. In this way, each ratio abundance represents the yield of formation of Pi to the precursor ion to the applied CV. The MS/MS analyzes were acquired in multiple reaction monitoring (MRM) by selecting, from the ERMS experiments, the most characteristic ion transitions for each analyte (Table 3).

In order to evaluate the human plasma chemical stability of studied compounds, a dedicate MRM method was arranged for each pair of isomers with the specific MS/MS transitions of the IS and the monitored isomers, each acquired for 25 ms (dwell time).

### 4.5. Sample Preparation

Each sample was prepared by adding 10 µL of proper spiked solution to 100 µL of human plasma matrix in a 1.5 mL microcentrifuge tube. The obtained solution corresponds to 1 μM of analyte that undergo degradation study. Each set of samples was prepared in triplicate and incubated for four different times: 0, 30, 60, and 120 min at 37 °C. Therefore, each stability panel of studied analyte was represented by a batch of 12 samples (4 incubation times, 3 replicates). After the incubation, 300 µL of IS solution was added to the sample and then centrifuged (room temperature for 5 min at 800× *g*). Then, the supernatant was transferred to autosampler vials, dried under a gentle stream of nitrogen and dissolved in 1.0 mL of 10 mM of formic acid in mQ water:acetonitrile 80:20 solution [23,24,25,27,28,29,30]. Each sample batch included the blank samples of human plasma matrix, prepared as described above, with the addition only of the IS solution. Thus, the analysis of the blank samples can check any interference in the analyte MRM signals due to the matrix components. Three sets of six replicates for each analyte were prepared to evaluate the matrix effect (ME) and the analyte recovery (RE) of the proposed method [31]. The same evaluation was extended to the IS to verify its reliability as a quantitative reference. Set A was prepared by mixing 10 µL of spiked solution with 300 µL of IS solution. Set B was obtained by mixing 100 µL of plasma with 300 µL of acetonitrile and, after centrifugation and separation of the supernatant, by adding 10 µL of spiked solution to each analyte or 300 µL of IS solution. Set C, in contrast, was prepared by mixing 10 µL of spiked solution with 300 µL of IS solution and 100 µL of plasma, then centrifuging and collecting the supernatant. The obtained solutions were transferred in autosampler vials, dried under a gentle nitrogen stream and dissolved in 1 mL of 10 mM of formic acid solution in mQ water:acetonitrile 80:20. Following the procedure described above, the expected concentrations of the samples (degradation, ME and RE sets) ranged between 50–60 ng mL^−1^ (depending on the MW of the studied compound), values placed in the middle of calibration curves. The final solutions of blank and all of the sets of samples, prepared as described above, were analyzed using the proper HPLC-MS/MS method.

### 4.6. Validation of HPLC-MS/MS Methods

Calibration curves of analytes were obtained by plotting the peak area ratios (PAR), calculated between the analyte and the IS quantitation ions signal, versus the nominal concentration of the calibration solution. A linear regression analysis was applied to obtain the best fitting function between the calibration points. Taking into account that both the calibration and sample solutions were prepared with the same procedure, the calibration curve parameters, the LOD and LOQ values for each analyte, referred to the concentration of sample solution actually analyzed by the HPLC-MS/MS methods. In order to obtain reliable LOD and LOQ values, the standard error (SE) of response and slope approach was employed [32]. The estimated SEs of the responses of each analyte were obtained by the SE of y-intercepts (Y-SE) of regression lines and the elaboration of data was obtained from the HPLC-MS/MS analyzes of calibration solutions [33]. The ME and RE values were determined by comparing the absolute peak areas of analytes of the three sets of samples (set A, B, and C), prepared as described in Section 2.6, following the formulas shown below [31]:(2)ME (%)= BA×100
(3)RE (%)=CB×100

### 4.7. LEDA Algorithm

In the proposed HPLC conditions, the chromatographic peaks of analytes were unresolved; therefore, LEDA algorithm application was necessary to elaborate the peak area of each MRM channel of the unresolved isomer signals, separating their components and assigning the correct abundance to the identified isomers. The LEDA algorithm is a matrix composed by *n* linear equations (see Equation (1)) to allow the resolution of the mixtures of *n* isomers. Naturally, to increase the specificity and reliability of isomers speciation, an overdetermined system of linear equations could be assembled; in this study, the LEDA matrix was composed by a number >*n* of linear equations (see Equation (1)). All calculations for the deconvolution of MS/MS data are processed using an Excel™ macro. The deconvolution was performed by applying the algorithm either to the area abundances, obtained from the integrated peak intensities of each MRM channel, or to individual MS/MS data points of the chromatographic sample profile. In the first approach, the LEDA provides the relative amounts (%) of each know component present in the sample. In the second approach, each MRM signal is deconvoluted ‘scan-by-scan’ and assigned to the present isomers, allowing a graphical separation of the processed chromatographic profiles. The characteristic abundance ratios were calculated by data obtained from the highest level of the calibration curve (100 ng mL^−1^) of each pure isomer by the HPLC-MS/MS methods described above. The ratios between Pi vs. Ri selected in the MRM methods were calculated and the resulted values were reported in Table 4.

The LEDA tool performances were evaluated by processing the MS/MS data obtained from the analysis of the standard mixture samples, prepared in Section 4.3, and the results checked by comparison with expected values.

### 4.8. Chemical Stability Test

In general, the presence of an ester group in the structure of a new drug candidate compound leads to the performance of a plasma stability study in order to verify its possible hydrolysis by esterase enzymes. The plasma samples were prepared by spiking a known amount of compound (usually between 1 μM to 5 μM), then being analyzed by a proper method, and finally their stability profiles were plotted. The stability profile of each compound was obtained by monitoring the variation of the analyte concentration at different incubation times in plasma samples. Generally, when the substrate concentration was smaller than the Michaelis–Menten constant (K_M_), the enzymatic degradation rate is described as a first-order kinetics. Therefore, by converting the quantitative results as natural logarithm, the natural logarithm of concentrations can be plotted versus incubation times and, once the slope of the linear function is obtained, will represent the degradation rate constant (k). Then, the half-life (t_1/2_) of each tested compound can be easily calculated as follows:(4)t12=ln (0.5 μM)k

For the compounds that showed a value of k rate < 0.006 (ln(μM)/min.), a high t_1/2_ value will be determined. However, by considering the measure errors and that the highest incubation time involved in the study was 120 min, the t_1/2_ parameter of compounds with low k rate has been calculated up to the limit value of 120 min, and beyond has been indicated as >120 min. The hydrolytic activity of the employed pool of human plasma was checked by adding KEE (reference compound) and, after application of the same procedure followed for the studied compounds, its degradation plot (SF34) showed a decrease comparable to that found in literature (t_1/2_ ≤ 120 min.), confirming the activity of plasma enzymes [24,25,27,28]. 

## 5. Conclusions

In the proposed HPLC-MS/MS method, we have used a mathematical algorithm (LEDA) to solve and separate the MS/MS spectra from the unresolved chromatographic peak of a pair of isomers. The development of LEDA involved an investigation on the energetics of the fragmentation pathway and allowed for the selection of the better product ions for each analyte in terms of both sensitivity of detection and specificity, i.e., the capability to distinguish between isomeric compounds. It is noteworthy that the choice of characteristic product ions and the optimal abundance ratios play an important role for the application of the LEDA approach to a multi-component mixture analysis. The developed method was applied in a human plasma drug stability study of two couples of isomers to verify its effectiveness. The obtained results confirm the ability of the LEDA approach to distinguish the isomers, allowing the evaluation of their behavior in plasma samples. It is worth emphasizing that the investigation was carried out with a conventional HPLC-MS/MS system, without using expensive ultra-efficient chromatography systems or ancillary MS techniques (i.e., MS ion mobility). In the proposed approach, we have explored the potential of MS/MS techniques, introducing the “energetic dimension” of the experiment, which proved to be fruitfully employed to solve the MS problems in the recognition of isomers.

## Figures and Tables

**Figure 1 ijms-23-13139-f001:**
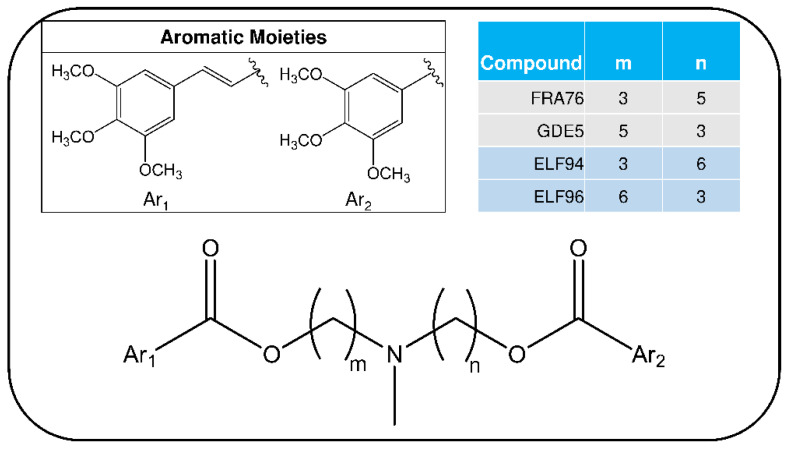
Chemical structures of P-gp inhibitors selected for this study.

**Figure 2 ijms-23-13139-f002:**
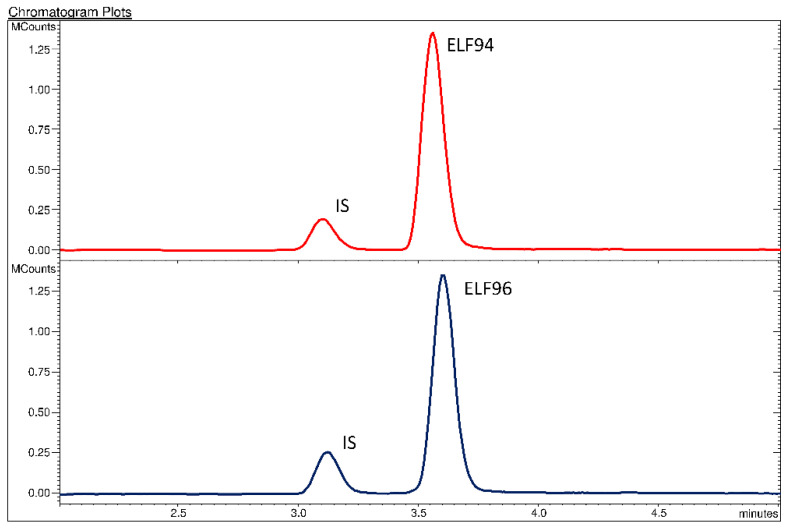
Chromatographic profiles of the HPLC-MS/MS analysis of the ELF94 and ELF96 isomers.

**Figure 3 ijms-23-13139-f003:**
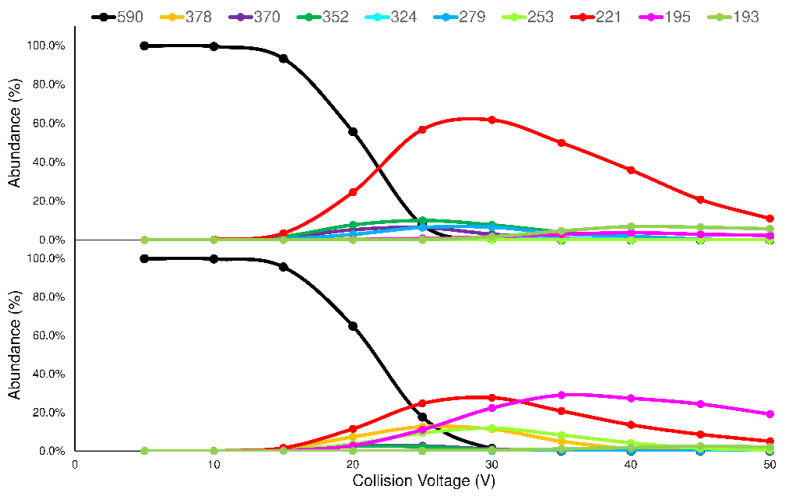
Comparison between the breakdown curves of selected MS/MS transitions of FRA76 (top) and GDE5 (bottom) isomers.

**Figure 4 ijms-23-13139-f004:**
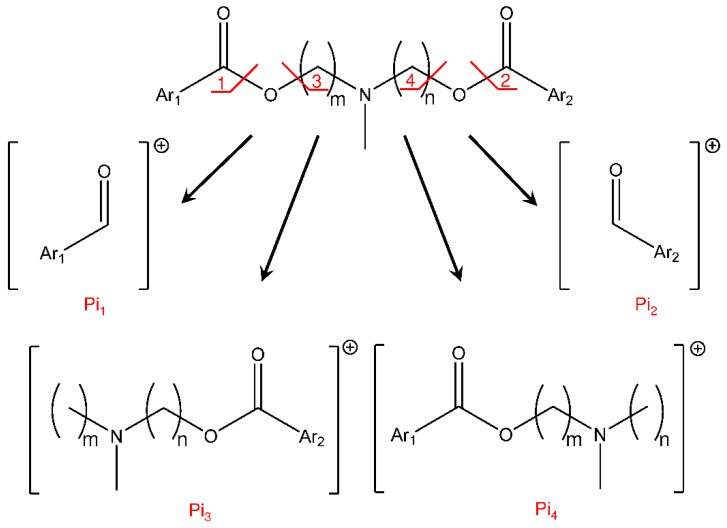
Common fragmentation pathways proposed for the studied P-gp inhibitors. Each bond cleavage site is marked by a red line and associated with a number, that is reported also to the related product ion (Pi).

**Figure 5 ijms-23-13139-f005:**
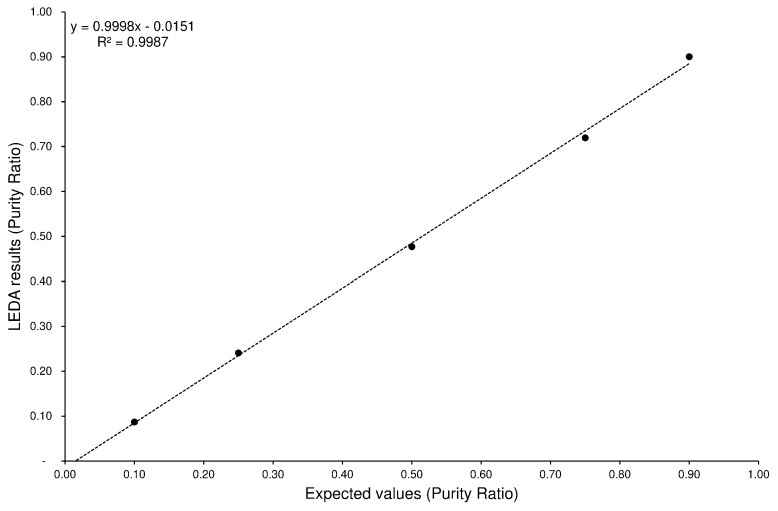
Validation plot obtained for the ELF96 isomer by processing the standard mixtures with LEDA_195–221_ matrix.

**Figure 6 ijms-23-13139-f006:**
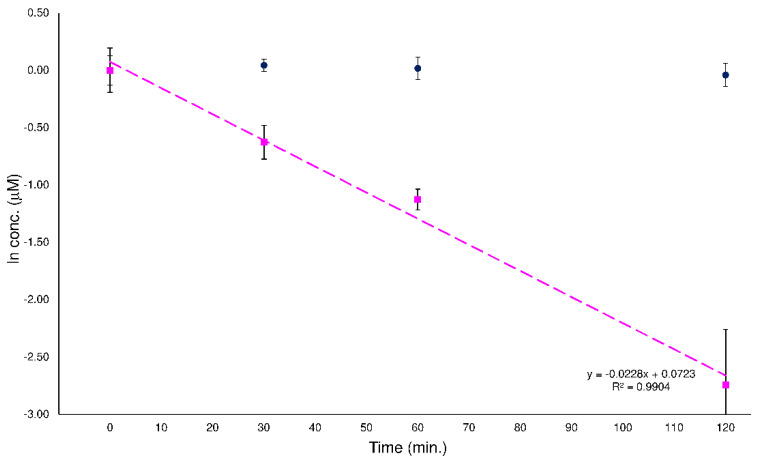
Degradation plots obtained by LEDA elaboration of the human plasma samples spiked with mixtures of the ELF94 (pink squares) and ELF96 (blue circles) isomers.

**Figure 7 ijms-23-13139-f007:**
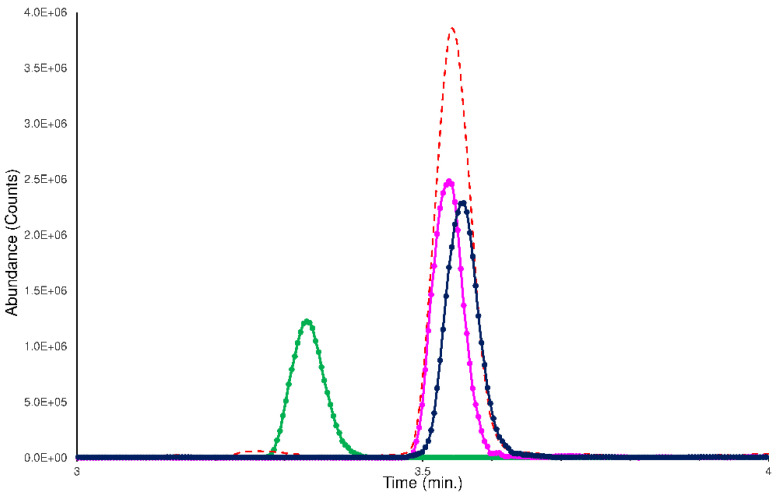
The LEDA reconstructed chromatographic profiles of the human plasma sample spiked with the mixture of the ELF94 (pink line) and ELF96 (blue line) isomers at the incubation time 0 min. The IS (green line) and Ri (red dotted line) signals were also reported.

**Table 1 ijms-23-13139-t001:** Validation plot parameters (slope, y-intercept, determination coefficient or R2 and standard error of linear function or SE-Lin) obtained by the elaboration of MS/MS data of standard mixtures of each pair of isomers by the LEDA matrices described above.

	Slope	y-Intercept	R^2^	SE-Lin
FRA76 LEDA_195–221_	1.05	−0.05	0.994	0.03
FRA76 LEDA_All_	1.05	−0.05	0.995	0.03
GDE5 LEDA_195–221_	1.03	0.01	0.996	0.03
GDE5 LEDA_All_	1.04	0.01	0.997	0.02
ELF94 LEDA_195–221_	1.02	−0.01	0.999	0.01
ELF94 LEDA_All_	1.02	−0.01	0.999	0.01
ELF96 LEDA_195–221_	1.00	−0.02	0.999	0.01
ELF96 LEDA_All_	1.01	−0.02	0.999	0.01

**Table 2 ijms-23-13139-t002:** The characteristic degradation parameters, half time (t_1/2_), degradation rate (k), with the estimated standard deviations (SD), calculated by conventional quantitative or LEDA processing of the data from the series of spiked human plasma samples.

	Pure Isomer t_1/2_ ± 2SD (min)	Pure Isomer k ± 2SD (ln(μM)/min)	Mix t_1/2_ ± 2SD (min)	Mix k ± 2SD (ln(μM)/min)
FRA76	43 ± 16	−0.017 ± 0.001	32 ± 18	−0.019 ± 0.003
GDE5	>120	<0.006	>120	<0.006
ELF94	49 ± 18	−0.014 ± 0.001	34 ± 14	−0.023 ± 0.005
ELF96	>120	<0.006	>120	<0.006

**Table 3 ijms-23-13139-t003:** MRM parameters used for acquisition of IS and each couple of isomers.

Compound	Precursor Ion (*m*/*z*)	MRM Signal	Product Ion (*m*/*z*)	CV (V)
IS	455		165	30
FRA76 GDE5	590	Ri	590 ^(^*^)^	10
		Pi_2_	195	35
		Pi_1_	221	25
		Pi_6_	253	30
		Pi_5_	279	30
		Pi_4_	352	25
		Pi_5_	378	25
ELF94 ELF96	604	Ri	604 ^(^*^)^	10
		Pi_2_	195	35
		Pi_1_	221	25
		Pi_6_	253	30
		Pi_5_	279	30
		Pi_4_	366	25
		Pi_5_	392	25

^(^*^)^ Ion transition used as reference ion signal (Ri) in the LEDA elaboration. MRM signal refers to the assigned number of the Ri or Pis ions reported in Section 2.2.

**Table 4 ijms-23-13139-t004:** Characteristic ion abundance ratios (Pi/Ri) ± standard deviation (SD) calculated by MS/MS data from 100 ng mL^−1^ solution of each pure isomer by HPLC-MS/MS methods described in Section 4.4.

Isomer Pair	Ratio Pi/Ri (*m/z*)	Isomer 1 Ratio Value ± SD	Isomer 2 Ratio Value ± SD
FRA76-GDE5	378/590	0.01 ± 0.01	0.26 ± 0.01
	352/590	0.20 ± 0.01	0.04 ± 0.01
	279/590	0.11 ± 0.01	0.01 ± 0.01
	253/590	0.01 ± 0.01	0.21 ± 0.01
	221/590	1.30 ± 0.02	0.60 ± 0.01
	195/590	0.09 ± 0.01	0.64 ± 0.03
ELF94-ELF96	392/604	0.01 ± 0.01	0.32 ± 0.01
	366/604	0.18 ± 0.01	0.01 ± 0.01
	279/604	0.08 ± 0.01	0.01 ± 0.01
	253/604	0.01 ± 0.01	0.18 ± 0.01
	221/604	1.10 ± 0.06	0.44 ± 0.03
	195/604	0.04 ± 0.01	0.70 ± 0.02

## Data Availability

Not applicable.

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
