# Peer review of "Simultaneous Degradation Study of Isomers in Human Plasma by HPLC-MS/MS and Application of LEDA Algorithm for Their Characterization"

_ijms, 2022, doi:10.3390/ijms232113139_

Round 1
Reviewer 1 Report
The authors report the application of energy-resolved tandem mass spectrometry coupled to HPLC to study mixtures of isomers in human plasma. They applied linear deconvolution algorithm to identify and quantify model mixtures of isomeric compounds in human plasma. This manuscript shows an interesting technique and might be published after the suggested corrections are done and adequate answers provided to the following questions:
In the abstract the authors claim their "... manuscript proposes a new tandem MS approach by introducing the energetic dimension of the experiment ...", but they also report a number of different papers related to similar energy-dependent MS/MS approaches in their introduction (line 55, p.2: "in the last 2 decades ..."). Probably the adjective "new" is overclaiming.
In the introduction, the authors do not describe accurately the different contributions and MS/MS methodologies already developped in the context of isomer distinction. It must be emphasized two groups of techniques: 1) single-valued MS/MS and, 2) energy-resolved MS/MS. Moreover, those techniques were applied in significantly different contexts which should also be more detailed and presented separately: a) to differentiate/identify mixtures of isomers and, b) to quantify a compound in the presence of an isomeric interference. The authors should be more accurate and emphasize in which respect their approach is novel and on which previously reported results they rely on. In particular, in the context of quantitative analysis, they may compare more precisely their approach with results from the groups of Leary et al. (Anal. Chem. 1999, 71, 19, 4142–4147, 10.1021/ac990553w) and the group of Memboeuf et al. (Anal. Chem. 2018, 90, 14126−14130, ) with which we can see strong convergences/similarities although they are not referenced in the manuscript. The authors may also compare, during their discussion (not only in the introduction), the performance of their approach in light of the results reported in the above 2 papers.
When looking at validation plots provided as supporting information (SF10 to SF17), it appears the LEDA algorithm is performing less at higher purity ratio, with a datapoint inducing strong bias due to lever effect (SF10 and SF11). SF12 and SF13 also exhibit a small curvature, which may more clearly appear when plotting residual analysis. The authors may comment on it and, evaluate if the restriction to a lower range of mixture ratio would improve the results.
SF18 and SF19 report MS/MS spectra reconstruction from their LEDA algorithm. The authors should provide the MS/MS spectra for pure compounds obtained in the same experimental conditions, to support a sound discussion on this point.
Linear regression results in SF20 and SF21 do not compare very well with data from LEDA presented in SF22. Why ? and how much difference can be afforded/explained ? Idem with SF23 and SF24 to be compared with SF25.
In SF26 to SF33 reporting the LEDA reconstructed chromatograms for the mixtures of the different isomers, shall we not expect to find the chromatographic peaks for the two isomers to sum up and correspond to Ri chromatographic data (as it corresponds to the precursor ion intensities) ? Can the authors explain why it is not the case ?
In SF32, the FRA76 chromatogram actually shows 2 peaks. Does it have a meaning ?
The degradation plot of human plasma spiked with KEE reference is quite surprising, as the authors represent a linear regression with 2 datapoints ... ? Probably some datapoints are missing, aren't they ?
The verapamil standard is not correctly described (which isomer was used ? or racemic mixture ?), the authors may provide the CAS number. The authors should argue in more details on the relevance of using verapamil as an internal standard in their work. The chemical structure is quite different to the compounds analyzed. They should give more arguments on its suitability to be used as a standard in this case.
Author Response
Dear Editor,
I received the Reviewers’ comments to the manuscript " Simultaneous degradation study of isomers in human plasma by HPLC-MS/MS and application of LEDA algorithm for their characterization" and I wish to thank for their useful suggestions.
Enclosed please find the revised version of the manuscript amended according to their suggestions.
A point wise list of response to the comments and the corrections (red text), now enclosed in the paper, is attached to this message.
I hope that in present form the paper could be considered worthy of publication.
Thank you for the opportunity you gave us to resubmit the paper.
Waiting for again kind answer, I remain
Sincerely yours
Gianluca Bartolucci, (corresponding Author)
Response to Reviewer 1
Comments
The authors report the application of energy-resolved tandem mass spectrometry coupled to HPLC to study mixtures of isomers in human plasma. They applied linear deconvolution algorithm to identify and quantify model mixtures of isomeric compounds in human plasma. This manuscript shows an interesting technique and might be published after the suggested corrections are done and adequate answers provided to the following questions:
In the abstract the authors claim their "... manuscript proposes a new tandem MS approach by introducing the energetic dimension of the experiment ...", but they also report a number of different papers related to similar energy-dependent MS/MS approaches in their introduction (line 55, p.2: "in the last 2 decades ..."). Probably the adjective "new" is overclaiming.
The authors presented "MS/MS approach" such as a new “way” respect to the conventional methodological approaches commonly reported in a plethora of MS/MS devoted papers. In the "new approach" should be included the entire concept of the application; then as reported in the abstract, "The chromatographic set up (HPLC) was tuned to minimize the run time, without requiring high efficiency or resolution between the isomers. Then, the MS/MS properties were explored to solve the signal assignment, by performing a series of energy resolved experiments to optimize the parameters and applying an interesting post-processing data elaboration tool (LEDA)." The authors believe that the adjective "new" together with "approach " can be suitable to describe the purpose in the abstract. On the other hand, none of the cited bibliography reports a similar comprehensive HPLC-MS/MS.
In the introduction, the authors do not describe accurately the different contributions and MS/MS methodologies already developped in the context of isomer distinction. It must be emphasized two groups of techniques: 1) single-valued MS/MS and, 2) energy-resolved MS/MS. Moreover, those techniques were applied in significantly different contexts which should also be more detailed and presented separately: a) to differentiate/identify mixtures of isomers and, b) to quantify a compound in the presence of an isomeric interference. The authors should be more accurate and emphasize in which respect their approach is novel and on which previously reported results they rely on. In particular, in the context of quantitative analysis, they may compare more precisely their approach with results from the groups of Leary et al. (Anal. Chem. 1999, 71, 19, 4142–4147, 10.1021/ac990553w) and the group of Memboeuf et al. (Anal. Chem. 2018, 90, 14126−14130,10.1021/acs.analchem.8b05016) with which we can see strong convergences/similarities although they are not referenced in the manuscript. The authors may also compare, during their discussion (not only in the introduction), the performance of their approach in light of the results reported in the above 2 papers.
The authors are aware of the many MS/MS strategies developed in the isomers recognition. Some of these are not mentioned because they are out of scope of the study. In fact, we reported among our aims the sentence: ".....to carry out an adequate specificity to distinguish the isomers in the sample without the support of any structural manipulation (i.e. derivatization and isotopic enrichment) or chromatographic separation." e.g. Leary et al. Anal. Chem. 1999, 71, 19, 4142–4147, 10.1021/ac990553w has not been reported since is based on the derivatization of the analytes, approach excluded from the purposes of the article. On the other hand, this manuscript does not want to examine previous works nor make comparisons. The purpose is to present properly our investigation; the comparison between the different approaches to emphasize convergences/similarities will be task of the authors of an upcoming review. Unfortunately, we had missed the article of Memboeuf et al. (Anal. Chem. 2018, 90, 14126−14130,10.1021/acs.analchem.8b05016) that now was included in the references: reference 16. Furthermore, to emphasize the different groups of selected MS/MS techniques, the authors have added in the manuscript the sentence as follows: " Among of these strategies, the most promising and interesting for a widespread application in isomers recognition are based on (1) energy-resolved tandem mass experiments [12-19] and (2) kinetics of the ion-molecule interaction [20-22], even if the latter are limited to the use of the ion trap."
When looking at validation plots provided as supporting information (SF10 to SF17), it appears the LEDA algorithm is performing less at higher purity ratio, with a datapoint inducing strong bias due to lever effect (SF10 and SF11). SF12 and SF13 also exhibit a small curvature, which may more clearly appear when plotting residual analysis. The authors may comment on it and, evaluate if the restriction to a lower range of mixture ratio would improve the results.
As the authors reported on the manuscript (“…..a tool should be introduced that describes the accuracy and precision of the method considering all the processed solutions.”), the validation plot should be a representation of the accuracy and precision abilities of the method in a range of concentration levels tested. Therefore, the evaluation of the parameters of calculated linear function depict the quali-quantitative performance of the proposed method, e.g. a single value of accuracy (slope of the linear function) will characterize the entire concentrations range of analyte in tested mixtures. Hence the validation plot does not used as calibration function but as evaluator of method performances in a studied range of analyte concentration. On the contrary, too many values must be reported to describe a single parameter; e.g. each analyte (at least two isomers), repeated for each of tested concentration level, for each parameter evaluated (accuracy, precision, etc…) a series of values should be arranged (a analite x b parameter evaluated x c concentration levels = a x b x c values calculated to evaluate the performances of the method).
SF18 and SF19 report MS/MS spectra reconstruction from their LEDA algorithm. The authors should provide the MS/MS spectra for pure compounds obtained in the same experimental conditions, to support a sound discussion on this point.
The authors reported in the figures SF18 and SF19 the extracted MRM spectra of pure compounds obtained in the same experimental conditions of the mixture samples. Since, different CV are applied on each single SRM transition, the authors reported the sentence “Reconstructed MS/MS spectra…”. However, the authors agree with the reviewer that the sentence is confused and rephrased the SF18 and SF19 figure captions as follows: “Extracted MRM spectra of the….”
Linear regression results in SF20 and SF21 do not compare very well with data from LEDA presented in SF22. Why ? and how much difference can be afforded/explained ? Idem with SF23 and SF24 to be compared with SF25.
These plots descried the behavior of the isomers in the human plasma sample during the incubation time at 37 °C as reported in the manuscript in the section 2.6 and the results in table 2. The plots related to the isomers GDE5 and ELF96 none linear regressions were calculated because none significant variation of their concentration were registered. Therefore, the related plots are comparable. Concerning the substrate isomers (FRA76 and ELF94) were applied a linear regression function to evaluate their rate of degradation. Taking into the account that the compared k values were obtained from different sample batches of human plasma, the authors believe that the results are comparable. In fact, we comment in the manuscript as follows: “The obtained results confirmed the hydrolysis of substrate-isomers (FRA76 and ELF94) by plasma enzymes, while the concentration of stable isomers (GDE5 and ELF96) remained constant in all sample sets tested. The comparison of the degradation parameters (t1/2 and k values) reported in Table 2 highlights some differences between the absolute values of the data obtained from the sets of samples (i.e. FRA76 t1/2 43 and 32 min.), but these differences are statistically non-significant (considering the error values, the confidence range of the measurements overlaps).”
In SF26 to SF33 reporting the LEDA reconstructed chromatograms for the mixtures of the different isomers, shall we not expect to find the chromatographic peaks for the two isomers to sum up and correspond to Ri chromatographic data (as it corresponds to the precursor ion intensities) ? Can the authors explain why it is not the case ?
The authors reported the accuracy and precision of the method in section 2.4 (Table 1) of the manuscript and in the reconstructed chromatograms the intensities of the isomer peaks respect these accuracy and precision values.
In SF32, the FRA76 chromatogram actually shows 2 peaks. Does it have a meaning ?
Also in this case, it should consider the errors of the method and, considering the low concentration of the residual isomer, it could happen a peak deformation. However, this problem does not affects the results, respecting the quantitative performances reported in Table 2.
The degradation plot of human plasma spiked with KEE reference is quite surprising, as the authors represent a linear regression with 2 datapoints ... ? Probably some datapoints are missing, aren't they ?
The authors report in the manuscript (section 4.8) that the KEE is used as reference compound to verify the enzyme activity of human plasma batch. Its degradation after two hours of incubation is sufficient to assure the enzyme activity. The linear function calculated and reported in figure SF34 should be considered only to report an indicative t1/2 in the manuscript.
The verapamil standard is not correctly described (which isomer was used ? or racemic mixture ?), the authors may provide the CAS number. The authors should argue in more details on the relevance of using verapamil as an internal standard in their work. The chemical structure is quite different to the compounds analyzed. They should give more arguments on its suitability to be used as a standard in this case.
The authors agree with the reviewer and correct the quality descriptor of the verapamil hydrochloride, but also for Ketoprofen, in the manuscript with "racemic analytical standard". We think that, in this case, the CAS numbers should be needless. About the structure differences with the analytes we are aware for it, but this is the better compromise. Taking into the account that the analytes are designed and synthesized referring of the verapamil structure (leading compound). However, in the manuscript we checked its suitability as internal standard in section 2.5 and report: "Finally, the obtained ME and RE values for the IS resulted about 100 % for the tested matrix, confirming that it was an appropriate choice to monitor the studied compounds."

Reviewer 2 Report
The manuscript is about a novel HPLC-MS/MS method using a mathematical algorithm (LEDA) to solve and separate the MS/MS spectra from the unresolved chromatographic peak of a pair of isomers. The manuscript is well-written, and I really like the idea of the LEDA algorithm to use in quantitation. The manuscript is well-worth publishing. I have only a few minor comments:
- - Chromatographic efficiency (N) is expressed as the number of theoretical plates, the Authors should mention it in the manuscript.
- - Page 4, line 130 there is an unnecessary dot in the title. It also can be found in 4.1 and 4.2
Author Response
Dear Editor,
I received the Reviewers’ comments to the manuscript " Simultaneous degradation study of isomers in human plasma by HPLC-MS/MS and application of LEDA algorithm for their characterization" and I wish to thank for their useful suggestions.
Enclosed please find the revised version of the manuscript amended according to their suggestions.
A point wise list of response to the comments and the corrections (red text), now enclosed in the paper, is attached to this message.
I hope that in present form the paper could be considered worthy of publication.
Thank you for the opportunity you gave us to resubmit the paper.
Waiting for again kind answer, I remain
Sincerely yours
Gianluca Bartolucci, (corresponding Author)
Response to Reviewer 2
Comments
Dear Authors,
The manuscript entitled "Simultaneous degradation study of isomers in human plasma by HPLC-MS/MS and application of LEDA algorithm for their characterization" by Marco Pallecchi , Laura Braconi , Marta Menicatti , Sara Giachetti , Silvia Dei , Elisabetta Teodori and Gianluca Bartolucci has been submitted to IJMS in Molecular Biophysics section.
The manuscript shows a new approach in the isomer recognition using LC-MS/MS and LEDA algorithm in human plasma samples. A quite a similar approach has been proposed by these authors before this year and published in Journal of Pharmaceutical and Biomedical Analysis.
Although, the manuscript is well written, the results are clearly presented and there is undoubtedly a novelty issue with using LEDA algorithm I have some struggle concerning the concept itself. How helpful it would be for other scientists?
The authors propose this approach to simplify the setup of the isomers recognition methods. Since the knowing poor specifity of the MS/MS in this field, it can be a new way of overcoming this issue.
Would they prefer to use this instead of traditional sample preparation?
In the conventional methods the isomers are chromatographically separated and, to achieve their resolution, a long time to setup is required. Furthermore, the optimized method generally is effective only for a pair of isomers. In this study, we have worked with mixtures of isomers by using the same chromatographic method, tuned to have short runs (short column and rapid elution gradient), and developed the MS/MS specificity to allow the isomers recognition.
As we all know the MS/MS methods have poor specificity in the recognition of isomers therefore this could be a new way of overcoming this issue.
The authors hope that the proposed approach is used by the scientific community to allow its development and spreading.
All in all I accept the manuscript after minor revision.

Reviewer 3 Report
Dear Authors,
The manuscript entitled "Simultaneous degradation study of isomers in human plasma by HPLC-MS/MS and application of LEDA algorithm for their characterization" by Marco Pallecchi , Laura Braconi , Marta Menicatti , Sara Giachetti , Silvia Dei , Elisabetta Teodori and Gianluca Bartolucci has been submitted to IJMS in Molecular Biophysics section.
The manuscript shows a new approach in the isomer recognition using LC-MS/MS and LEDA algorithm in human plasma samples. A quite a similar approach has been proposed by these authors before this year and published in Journal of Pharmaceutical and Biomedical Analysis.
Although, the manuscript is well written, the results are clearly presented and there is undoubtedly a novelty issue with using LEDA algorithm I have some struggle concerning the concept itself. How helpful it would be for other scientists? Would they prefer to use this instead of traditional sample preparation? As we all know the MS/MS methods have poor specificity in the recognition of isomers therefore this could be a new way of overcoming this issue.
All in all I accept the manuscript after minor revision.
Author Response
Dear Editor,
I received the Reviewers’ comments to the manuscript " Simultaneous degradation study of isomers in human plasma by HPLC-MS/MS and application of LEDA algorithm for their characterization" and I wish to thank for their useful suggestions.
Enclosed please find the revised version of the manuscript amended according to their suggestions.
A point wise list of response to the comments and the corrections (red text), now enclosed in the paper, is attached to this message.
I hope that in present form the paper could be considered worthy of publication.
Thank you for the opportunity you gave us to resubmit the paper.
Waiting for again kind answer, I remain
Sincerely yours
Gianluca Bartolucci, (corresponding Author)
Response to Reviewer 3
Comments
The manuscript is about a novel HPLC-MS/MS method using a mathematical algorithm (LEDA) to solve and separate the MS/MS spectra from the unresolved chromatographic peak of a pair of isomers. The manuscript is well-written, and I really like the idea of the LEDA algorithm to use in quantitation. The manuscript is well-worth publishing. I have only a few minor comments:
- - Chromatographic efficiency (N) is expressed as the number of theoretical plates, the Authors should mention it in the manuscript.
The authors added in the text as follows: “The proposed HPLC method achieved a poor efficiency, evaluated as the number of theoretical plates (N ≈ 5000), mainly due to the short column used.”
- - Page 4, line 130 there is an unnecessary dot in the title. It also can be found in 4.1 and 4.2
The authors correct the numbers of sections in whole of document.

Round 2
Reviewer 1 Report
I do not see more to comment on the work done by the authors, except that the first sentence of abstract:
"The manuscript proposes a new tandem mass spectrometry (MS/MS) approach in the isomer recognition by introducing the "energetic dimension" of the experiment."
does not mean what the authors explain in their rebuttal letter. The novelty of their work is not on the "introduction of energetic dimension in MSMS" for isomer recognition, which has been done for several decades already. More accurate phrasing is necessary in this very important section of the paper.
Other corrections/answers are fine.
Author Response
Response to Reviewer 1
Comments
I do not see more to comment on the work done by the authors, except that the first sentence of abstract:
"The manuscript proposes a new tandem mass spectrometry (MS/MS) approach in the isomer recognition by introducing the "energetic dimension" of the experiment."
does not mean what the authors explain in their rebuttal letter. The novelty of their work is not on the "introduction of energetic dimension in MSMS" for isomer recognition, which has been done for several decades already. More accurate phrasing is necessary in this very important section of the paper.
Other corrections/answers are fine.
The authors, in agreeent with reviewer’s suggestion, propose: “The manuscript proposes a tandem mass spectrometry (MS/MS) approach in the isomer recognition by playing in the "energetic dimension" of the experiment.”
